# CAML-PSPNet: A Medical Image Segmentation Network Based on Coordinate Attention and a Mixed Loss Function

**DOI:** 10.3390/s25041117

**Published:** 2025-02-12

**Authors:** Yuxia Li, Peng Li, Hailing Wang, Xiaomei Gong, Zhijun Fang

**Affiliations:** 1School of Electronic and Electrical Engineering, Shanghai University of Engineering Science, Shanghai 201620, China; sueslyx@gmail.com (Y.L.); 02200009@sues.edu.cn (H.W.); 2Department of Radiation Oncology, Shanghai Pulmonary Hospital, Tongji University School of Medicine, Shanghai 200433, China; 18701837650@126.com

**Keywords:** lung tumor, PSPNet, attention mechanism, mixed loss, image segmentation

## Abstract

The problems of missed segmentation with fuzzy boundaries of segmented regions and small regions are common in segmentation tasks, and greatly decrease the accuracy of clinicians’ diagnosis. For this, a new network based on PSPNet, using a coordinate attention mechanism and a mixed loss function for segmentation (CAML-PSPNet), is proposed. Firstly, the coordinate attention module splits the input feature map into horizontal and vertical directions to locate the edge position of the segmentation target. Then, a Mixed Loss function (MLF) is introduced in the model training stage to solve the problem of the low accuracy of small-target tumor segmentation. Finally, the lightweight MobilenetV2 is utilized in backbone feature extraction, which largely reduces the model’s parameter count and enhances computation speed. Three datasets—PrivateLT, Kvasir-SEG and ISIC 2017—are selected for the experimental part, and the experimental results demonstrate significant enhancements in both visual effects and evaluation metrics for the segmentation achieved by CAML-PSPNet. Compared with Deeplabv3, HrNet, U-Net and PSPNet networks, the average intersection rates of CAML-PSPNet are increased by 2.84%, 3.1%, 5.4% and 3.08% on lung cancer data, 7.54%, 3.1%, 5.91% and 8.78% on Kvasir-SEG data, and 1.97%, 0.71%, 3.83% and 0.78% on ISIC 2017 data, respectively. When compared to other methods, CAML-PSPNet has the greatest similarity with the gold standard in boundary segmentation, and effectively enhances the segmentation accuracy for small targets.

## 1. Introduction

Medical images are generally characterized by low contrast and difficulties in identify the boundaries between different tissues; these medical images can be presented across multiple modalities. The intrinsic complexity of medical images, often marked by low contrast and the challenging task of distinguishing tissue boundaries, necessitates advanced analytical approaches. Bridging this gap, the evolution of deep learning has provided a transformative motivation for the field. Recently, deep learning has been rapidly developed and different modalities of medical image segmentation tasks have shown good results by deep learning methods. Empirical evidence from an array of studies suggests that deep learning techniques surpass traditional machine learning techniques in yielding more dependable outcomes for medical image segmentation tasks [1,2,3].

Currently, several network architectures have been at the forefront of the field. The Fully Convolutional Network (FCN) [4] stands out for its unique adaptation in medical imaging, where it employs fully convolutional layers in lieu of fully connected ones. This design is capable of processing images of varying sizes as input and producing segmentation results with matching dimensions. FCN leverages convolution and multi-scale information fusion techniques to enhance segmentation precision while preserving detailed image features. Another notable architecture is SegNet [5], which operates on an encoder–decoder framework and performs feature extraction and pixel-level classification. SegNet introduces a novel approach to pooling by using maximal pooling indices, which serves to decrease the model’s computational burden. Parallel to these developments is U-Net [6], a model specifically tailored for the segmentation of biomedical images. It boasts a symmetric U-shaped structure that adeptly combines high-resolution pathways with a contracting path to deliver highly accurate segmentation results. Building on the FCN framework, the DeepLab network [7] also caters to biomedical image segmentation. It enhances the basic FCN structure by incorporating atrous convolution, which allows it to manage the segmentation of images at varying scales effectively. Although CNN architectures have advanced considerably in segmentation, there remains the challenge of detail loss during the pooling and convolution operations, which can compromise the quality of the segmentation outcomes. The continuous down-sampling process inherent in these networks tends to obscure finer details, which is a critical limitation when high precision is required, such as in medical image analysis. Thus, ongoing research is directed toward refining these networks to maintain detail integrity while achieving the overarching goal of accurate and high-resolution segmentation.

The Pyramid Scene Parsing Network (PSPNet) [8] has garnered attention for its novel approach to semantic segmentation. It integrates a pyramid pooling module that aggregates contextual information across varying scales and synthesizes multi-scale features to enhance segmentation accuracy. Initially applied to scene parsing, PSPNet assigns category labels to each pixel while capturing information on category, location, and shape. Its capability to handle complex issues such as contextual relationship mismatches, category confusion, and obscurity is notable. Furthermore, the network’s proficiency in fusing global and local feature sets via pyramid pooling operations significantly mitigates the issue of feature detail loss. This makes PSPNet a formidable contender in semantic segmentation tasks. Wu et al. [9] introduced a depth-guided fusion module on the basis of PSPNet and used a channel attention module during the network’s feature extraction stage, and obtained a strong advantage in object edge recognition and segmentation and obtained effective results. Zhu et al. [10] introduced an improved network based on a PSPNet network to segment coronary angiography images, achieving a segmentation accuracy of 0.957.

Connecting these advancements with the broader scope of attention mechanisms [11] in neural networks, it becomes clear that the ability to concentrate on pertinent segments of input data is integral to the improvement of network performance and generalization. A traditional attention mechanism is based on content, and the attention weight is computed based on the similarity, which cannot obtain the target position and object edge information well. The Coordinate Attention (CA) mechanism [12] is used in sequence modeling; it emphasizes the positional information within the sequence, enabling the model to capture relationships between various positions.

Transitioning to the topic of model training [13], selecting an appropriate loss function is crucial. The chosen loss function quantifies the deviation between the model’s predictions and the ground truth values, moving toward greater predictive accuracy through learning and optimization. Usually, loss functions have the role of model evaluation, back propagation and goal optimization in the process of model training. However, in some cases, due to data imbalance and small objectives, a mixed loss function combining different loss functions is needed to improve the model training, which optimizes the model training effect to a large extent.

Addressing the issues related to small targets—missed segmentation and wrong segmentation due to blurred tissue boundaries in the task of medical image segmentation in actual clinics—this study introduces PSPNet for natural scene parsing in the task of medical image segmentation and proposes a novel network architecture, CA mechanism and mixed loss function based on the PSPNet network. The approach first employs the CA mechanism to pinpoint the tumor region, followed by the introduction of the mixed loss function to expedite model convergence, which can effectively segment tumors. This paper makes the three contributions:(1)A novel network architecture, CAML-PSPNet, utilizes the coordinate attention module during network feature extraction to accurately locate the feature region and effectively assist the network in accurate segmentation.(2)Addressing the limitations of existing hybrid loss functions in segmenting small-target tumors, a new Mixed Loss is proposed; this expedites the model’s convergence rate while effectively improving the accuracy of segmentation by defining new scale coefficients.(3)MobileNetV2 serves as the backbone network in the PSPNet network, which greatly reduces the computation time and parameter count.

## 2. Related Works

### 2.1. CNN-Based Segmentation

Deep learning-powered segmentation algorithms, particularly CNNs, have become the vanguard, obviating the need for manual feature design by automating feature extraction and task-specific segmentation [14,15,16]. The Fully Convolutional Network (FCN) revolutionizes CNNs by introducing convolutional layers in place of fully connected ones and employing transposed convolution for pixel-wise classification, while U-Net further refines this by integrating more up-sampling and skip connections, significantly enhancing segmentation precision and achieving wide success in medical applications.

As deep learning progresses steadily, with Zhang et al. [17] having introduced SAU-Net, excellent experimental results have been obtained in brain tumor datasets. Wu et al. [18] introduced a segment-anything network (MSA) adapted to medical images, which is able to perform outstandingly in 19 medical image segmentation tasks, surpassing the current leading medical image segmentation networks. Zhao et al. [19] introduced DSU-Net, which clearly considers the information of the fuzzy region and applies the proposed distraction module to the process of local information extraction, which is more effective than other segmentation networks in lung tumor segmentation. Furthermore, Yang et al. [20] introduced an efficient 3D U-Net network featuring the ResNet architecture, which is capable of learning richer global and local lung tumor representations, and obtains a segmentation Dice accuracy of 0.691 on the TCIA dataset.

Transformer architecture has been applied to computer vision tasks and combined with CNN networks, and has been introduced to medical image segmentation tasks, achieving outstanding results [21,22,23,24]. In particular, Wang et al. [25] combined CNN and a transformer structure, with experiments on the BraTS 2020 and 2019 test sets achieving the best segmentation results.

### 2.2. Segmentation Based on the Attention Mechanism

The attention mechanism draws on the thinking mode of human visual signal processing [26]. Attentional mechanisms capture dependencies of remote information and are widely used in computer vision, which allows models to selectively concentrate on various sections of the input sequence when performing sequence modeling or processing.

In recent years, researchers have persistently integrated the attention mechanism into medical image segmentation networks. Oktay et al. [27] introduced Attention Gate, which filters the output of the U-Net down-sampling process through the AG and then connects it to the up-sampling result in order to eliminate noise and irrelevant information in the hopping process, and to decrease the computational overhead of the model. Li et al. [28], inspired by Attention U-Net, proposed a connection-sensitive attention network for the task of accurate segmentation of retinal blood vessels. Qi et al. [29] included a self-attention module in the coding and decoding network and added residual connectivity to the regular convolution and segmented a stroke dataset. Cheng et al. [30] used masked attention in the transformer decoder, which was able to converge faster and improve performance, outperforming the current network in different segmentation datasets. The use of different attention mechanisms on different medical image segmentation tasks gives better segmentation results, especially in how the coordinate attention mechanism locates the target object position by dividing the feature map into horizontal and vertical directions, which allows the network to take into account inter-channel relationships as well as long-distance positional information in the image feature extraction stage, and by using pixel positional information, it allows the model to be more sensitive to the different location pixels, which is a great advantage in image segmentation tasks.

### 2.3. Segmentation Based on Hybrid Loss Function

Choosing the right loss function is very important for training the model and enhancing performance. And the design of a mixed loss function can combine the advantages of two or more loss functions to achieve better model performance and generalization ability. Mixed loss functions can come in the form of simple linear weighted combinations or more complex forms such as in applying different loss functions according to different stages or conditions.

Zulfiqar et al. [31] introduced a segmentation network, DRU-Net; the performance of the network is improved by the implemented hybrid loss function, which works well for the segmentation of pulmonary arteries. Zhang et al. [32] introduced a hybrid loss function in a task on the segmentation of veins in magnetically sensitive weighted imaging (SWI) which contains accurate classification and a hybrid loss function with a global region overlap term, and the best Dice coefficient value of 0.756 was obtained experimentally. Tan et al. [33] introduced a hybrid loss function to solve the fuzzy boundaries and heterogeneous pathological sensitivity problem in the task of segmenting the liver, and completed the Sliver07 challenge with a best score of 82.55. Zhang et al. [34] introduced a DFL − UNet + CBAM-based method for spot segmentation and disease identification, and the outcomes demonstrated superior segmentation results compared to other networks. Yousefirizi et al. [35] obtained the best performance in a lymphoma lesion segmentation task with a hybrid loss function incorporating distribution, region, and boundary loss components.

From the above research, it can be seen that deep learning segmentation models enable the automated segmentation of medical images. However, due to the problem that medical images exhibit multimodality, unclear organizational boundaries and different sizes of segmentation regions, a basic segmentation network cannot obtain effective segmentation results. In this paper, we chose the PSPNet network with multi-scale feature fusion, a pyramid pooling operation and a fusion of local and global features, and employ a coordinate attention mechanism instead of the conventional global attention mechanism; the number of parameters used is smaller, decreasing computational overhead. This reduction facilitates faster model inference times, which is crucial for clinical applications where rapid diagnosis is often required. The hybrid loss function is utilized to effectively address issues such as category imbalance and the inaccurate segmentation of small targets.

## 3. Methods

### 3.1. Architecture Overview

Semantic segmentation networks such as U-Net, Deeplab, and HrNet are commonly used. However, these networks lose the detailed information of the image after continuous pooling and convolutional operations, which affects the final segmentation accuracy. Initially, PSPNet was employed in the task of scene parsing for semantic segmentation, and, by using the pyramid pooling module, enhances the semantic information contained within the shallow features, so that the segmentation in the shallow layer has enough contextual information and the detailed information of the target’s. Meanwhile, PSPNet also adopts the atrous convolution module, which uses different atrous rates to expand the sensory field, capturing multi-scale contextual information, so that spatial accuracy can be effectively improved.

Although PSPNet uses a pyramid pooling operation to alleviate the problem of context information loss to some extent, the results of segmentation for PSPNet still fail to achieve the expected results due to the problems of small-target omission and organizing the fuzzy boundaries of target in the task of medical image segmentation. Based on this problem, this paper improves on the basis of a PSPNet network. Firstly, coordinate attention is introduced before and after the backbone feature extraction network to locate the position of the features, so that the model can more accurately focus on the significance of various positions, thus enhancing the model’s performance at critical locations, such as regions of interest and object edges. Secondly, a mixed loss function, Mixed Loss, is employed in the network to address the issue of data imbalance. Lastly, the ResNet backbone feature extraction network is substituted with MobilenetV2, to improve the model segmentation accuracy while decreasing the number of model parameters and saving computational resources. An overview of CAML-PSPNet is shown in Figure 1.

The network is segmented into four components: the input image, feature extraction, pyramid pooling and the segmentation result output. The input image enters into the backbone network MobilenetV2 to obtain a shallow feature map, and a coordinate attention mechanism is incorporated at both the beginning and end of this feature map, which calculates the weight of each feature point by introducing the position information, thus enabling enhanced focus on the local region of the entire image. Subsequently, the feature maps undergo pooling operations of various sizes within the pyramid pooling module, i.e., the input incoming feature layer is divided into 1 × 1, 2 × 2, 3 × 3, and 6 × 6 sub-regions, respectively, and then each sub-region is pooled on average. Finally, to ensure uniform size, the fused feature maps from the 1 × 1, 2 × 2, 3 × 3, and 6 × 6 layers are resized using up-sampling, and fused with the shallow features extracted from the backbone features, and the final segmentation results are derived following the convolution operation.

### 3.2. Coordinate Attention

Coordinate attention effectively incorporates positional information into channel attention, rendering it a novel and potent attention mechanism. The maps capture long-range dependencies along spatial directions, maintaining location details. By introducing the coordinate encoding and attention mechanism, coordinate attention enables the model to more effectively concentrate on various locations within the image and weight the features according to the importance of the location, thus improving the model’s capacity to perceive and represent specific locations, to achieve better performance in image processing tasks, as shown in Figure 2.

The coordinate attention mechanism involves two steps: coordinate attention generation and coordinate information embedding.

Coordinate information embedding: to effectively capture distant spatial interactions with accurate location details, the global average pooling is decomposed, and given the following input: F ∈ $R^{C\times W\times H}$. The feature maps are pooled in both the X- and Y-directions using two pooling kernels—(H, 1) and (1, W)—to generate the feature maps of C×H×1 and C×1×W. The feature maps are as follows:(1)$$\begin{array}{c} X:\;Z_{c}^{h}\left(h\right)=\frac{1}{W}\sum_{0\leq i\leq W}x_{c}\left(h,i\right)\;Z_{c}^{h}\in R^{C\times H\times 1} \end{array}$$(2)$$\begin{array}{c} Y:Z_{c}^{w}\left(w\right)=\frac{1}{H}\sum_{0\leq j\leq H}x_{c}\left(j,w\right)\;Z_{c}^{w}\in R^{C\times 1\times W} \end{array}$$

Coordinate attention generation: the embedded feature maps $Z_{c}^{h}$ and $Z_{c}^{w}$ are spliced along the spatial dimension after a 1 × 1 convolutional transformation and activated, as in Equation (3), followed by the separation of the two feature maps along the spatial dimensions, respectively, of $f^{h}\in R^{C/r\times H\times 1}\;and{\;f}^{w}\in R^{C/r\times 1\times W}$. The resulting attention vector is obtained by combination with the sigmoid activation functions, $g^{h}=\sigma \left(F_{n}\left(f^{h}\right)\right)\;and{\;g}^{w}=\sigma \left(F_{w}\left(f^{w}\right)\right)$. The final output formula of coordinate attention combines the input with the results from the X-direction and Y-direction operations, expressed as follows:(3)$$\begin{array}{c} f=\delta \left(F_{1}\left(\left[Z_{c}^{h},Z_{c}^{w}\right]\right)\right) \end{array}$$(4)$$\begin{array}{c} y_{c}\left(i,j\right)=x_{c}\left(i,j\right)\times g^{h}\left(i\right)\times {\;g}^{w}\left(j\right) \end{array}$$

In the formula, σ and δ denote the activation functions, respectively, and r is the reduction factor.

### 3.3. Mixed Loss Function

A mixed loss function is a way to use several different types of loss functions to comprehensively evaluate a model. Combining the Dice coefficient and Focal Loss enables the creation of a comprehensive hybrid loss function. The Dice coefficient quantifies the similarity of overlapping regions. The Dice coefficient loss function as follows:(5)$$\begin{array}{c} Dice\;Loss=1-\frac{2\times \sum_{i=1}^{N}G_{i}P_{i}}{\sum_{i=1}^{N}G_{i}+\sum_{i=1}^{N}P_{i}} \end{array}$$

Equation (5):${\;G}_{i}$ is the result of the *i*th pixel in the CT image undergoing manual segmentation by the expert;${\;P}_{i}$ is the network’s predicted result for the *i*th pixel to belong to the target class; N is the number of pixels in the segmented CT image. However, the Dice coefficient loss function is prone to bias for data with unbalanced categories, which tends to cause the model to focus excessively on large categories and ignore small categories. To solve this problem, Focal Loss is introduced:(6)$$\begin{array}{c} Focal\;Loss=-\alpha {\left(1-p\right)}^{\gamma }\log\left(p\right) \end{array}$$

Equation (6): α is the balancing factor; γ is the adjustment factor, p is the probability value predicted by the model. Focal Loss solves the problem of category imbalance with an adjustable hyperparameter, thereby enhancing the model’s categorization effect for a few categories.

Taking into full consideration the advantages of the two loss functions, the experimental process of this paper employs Mixed Loss:(7)$$\begin{array}{c} Mixed\;Loss=\lambda \times Dice\;Loss+\left(1-\lambda \right)\times Focal\;Loss \end{array}$$

The trade-off factor λ is used to balance the weights. By adjusting the value range of λ, the trade-off relationship between similarity and category imbalance can be adjusted. Because the target regions in medical images constitute a smaller portion of the overall image, in using the cross-entropy loss function, it is easy to be influenced by the background region, which reduces training efficiency. The mixed loss function is used to optimize the difficult-to-learn samples in the process of network backpropagation in a stable and targeted way, and the weight is reduced by Focal Loss, thus enabling the model to allocate greater attention to challenging samples and preserve intricate boundary details, and at the same time, the imbalance of pixel categories is solved by Dice Loss, which alleviates the noise caused by Focal Loss to a certain degree.

## 4. Experimental Results and Discussion

### 4.1. Datasets

The CT data for lung tumor segmentation used in the experiments are from Shanghai Pulmonary Hospital. To validate the capability of CAML-PSPNet in segmentation tasks, the segmentation performance is evaluated. Two publicly available datasets, the polyp segmentation data from colonoscopy cases, Kvasir-SEG [36], and the skin lesion segmentation data from dermatoscopy cases, ISIC 2017 [37], are selected for the comparison experiments. Table 1 shows the data details:

In the experimental data processing, firstly, to adapt the network requirements, the original medical image data were processed in different formats, the value of the background pixel point of the label of the original data, 0, and the value of the target pixel point, 255, were modified to 0 and 1, respectively. Secondly, the original sizes of the images of each dataset were different, and before inputting the data into the model, the image size was uniformly resized to 473 × 473 to fit the model training. Finally, data augmentation on the training dataset took place to improve the effect of training and improve the model generalization, which is mainly divided into the random flipping and rotating of the images, and, at the same time, the dataset is divided into the test set, validation set and training set, with a ratio of 1:1:8.

### 4.2. Experimental Environment and Parameter Settings

In this paper, the algorithm is implemented in the Python programming language and based on PyTorch 1.6.0. Meanwhile, the experiments are carried out under the Ubuntu 18.04.6 LTS operating system. The hardware for the experiments is the NVIDIA GeForce RTX 2080, which has 32 G of RAM. To optimize the model training process, the Stochastic Gradient Descent (SGD) algorithm is chosen. Moreover, the epoch of the experiment is set to 150 and the batch-size is set to 8.

### 4.3. Evaluation Indicators

In the image segmentation task, evaluation metrics are employed to quantify the similarity and accuracy. In order to quantitatively evaluate the segmentation result of the network, four common evaluation metrics are chosen: mean intersection over union (mIoU), mean pixel accuracy (mPA), accuracy and Recall.

Here, mIoU is used to evaluate the performance of semantic segmentation models:(8)$$\begin{array}{c} P_{mIOU}=\frac{N_{TP}}{N_{TP}+N_{FP}+N_{FN}} \end{array}$$

IoU calculates the degree of overlap between the predicted results and the real labels, which is used to measure the model’s localization accuracy for the target region.

The mPA is used for semantic segmentation:(9)$$\begin{array}{c} P_{mPA}=\frac{N_{TP}+N_{TN}}{N_{TP}+N_{FP}+N_{TN}+N_{FN}} \end{array}$$

The mPA indicates the number of pixels correctly predicted. In simple terms, it measures the overall accuracy for all pixel classifications.

Precision is a metric used to evaluate classification models:(10)$$\begin{array}{c} P_{precision}=\frac{N_{TP}}{N_{TP}+N_{FP}} \end{array}$$

Precision measures the proportion of samples predicted by the model to be in the positive category that are actually in the positive category.

Recall is a metric used to evaluate the performance of classification models and can also be used for semantic segmentation tasks:(11)$$\begin{array}{c} R_{recall}=\frac{N_{TP}}{N_{TP}+N_{FN}} \end{array}$$

Recall represents the number of samples correctly predicted by the model as positive cases as a proportion of the total number of actual positive class samples. It measures the model’s ability to detect the target class.

Here, N_TP_ denotes pixels predicted to be in the positive category and labeled in the positive category; N_TN_ denotes pixels predicted to be in the negative category and labeled in the negative category; N_FP_ denotes pixels predicted to be in the positive category and labeled in the negative category; N_FN_ denotes pixels predicted to be in the negative category but labeled in the positive category.

### 4.4. Results and Analysis

#### 4.4.1. PrivateLT Segmentation Experiments

The PrivateLT dataset contains 1008 CT images with different original image sizes, which are uniformly resized to 473 × 473. The proposed network CAML-PSPNet is experimentally compared with the currently popular Deeplabv3, HrNet, UNet and PSPNet networks. In Table 2, it can be seen that the proposed algorithm surpasses the other algorithms in four evaluation metrics. Its mIOU, mPA, Precision and Recall reach 73.2%, 85.19%, 83.87% and 85.19%, respectively, and the mIOU metrics are improved by 2.84%, 3.1%, 5.4% and 3.08%, respectively, compared to the Deeplabv3, HrNet, Unet and PSPNet networks. Comprehensive comparative analysis shows that the segmentation performance is better than other algorithms.

In Figure 3, the viewable area of the segmentation effect of PrivateLT is shown, and it can be seen that in the segmentation results of groups (a) and (b), the Deeplabv3, HrNet, UNet and PSPNet networks have the problems of unclear boundaries and large differences when compared with the labels, whereas the proposed CAML-PSPNet is able to more accurately segment the target, and it can obtain more consistent results in the segmentation of the tumor boundary, which proves the method is effective in solving the problem of unclear tumor boundaries. In the (c) group of segmentation visualization graphs, the Deeplabv3 network fails to segment the tumor location due to the small tumor region, the HrNet, UNet and PSPNet networks obviously have under-segmentation problems and the tumor edge segmentation is rougher, while CAML-PSPNet is able to segment the tumor region to a larger extent, indicating the effectiveness of the method for segmenting small targets. Compared with the existing popular segmentation models, the proposed algorithm has certain advantages.

#### 4.4.2. Kvasir-SEG Polyp Segmentation Experiments

Kvasir-SEG is a polyp dataset for medical image segmentation; 1001 colonoscopy images were selected and their size was processed to 473 × 473 dimensions for the experiment. Table 3 shows that the proposed algorithm has effective segmentation results when compared with other commonly used segmentation networks. In the visualization in Figure 4, polyps of different sizes, shapes and complexities, such as with blurred boundaries, can be accurately localized to obtain more precise segmentation results.

#### 4.4.3. ISIC 2017 Dermatology Segmentation Experiment

ISIC 2017 is a dermatologic dataset, and 960 dermoscopic images were selected for comparison experiments. Table 4 shows that the proposed algorithm surpasses other algorithms in the evaluation metrics of mIoU, mPA, Precision and Recall. From the segmentation visualization in Figure 5, the segmentation of the CAML-PSPNet network is closest to the labeled graph, especially at the edge of the tumor, where the segmentation effect is more accurate than other networks.

#### 4.4.4. Comparative Networks

In this research, the compared networks predominantly consist of fundamental networks such as Deeplabv3, HrNet, UNet and PSPNet. These networks have long exerted a substantial influence within the domain of segmentation. To further validate the effectiveness of the proposed network in the specialized field of medical image segmentation, a novel round of comparison is meticulously executed. Specifically, the networks Swin-UNETR and TransUNet, which have incorporated the transformer architecture in recent years, are selected for in-depth comparative experiments.

During these experiments, the mean intersection over mIOU and Precision are chosen as the key verification indicators. As presented in Table 5, it is evident that the proposed CAML-PSPNet network not only outperforms but also attains the optimal results in both the mIOU and Precision evaluation metrics. In the visual representation of the segmentation results, depicted in Figure 6, the segmentation effects of the two compared networks, in terms of mIOU and Precision, clearly exhibit discernible disparities when compared with those of the proposed network.

#### 4.4.5. Ablation Experiments

The proposed CAML-PSPNet algorithm network model is an improvement of the PSPNet network. To verify the effectiveness of using the coordinate attention mechanism and Mixed Loss for the segmentation task, ablation experiments were performed on PrivateLT. The PSPNet network was used as the baseline network to which no other modules were added. First, Dice Loss + Focal Loss (Mixed Loss) was tested in the PSPNet network, i.e., PSPNet+Mixed Loss; second, coordinate attention was introduced into the PSPNet for the experiments, i.e., PSPNet+Att; and lastly, the Mixed Loss function and coordinate attention were simultaneously applied to the PSPNet network for experimentation, i.e., CAML-PSPNet, the model proposed.

Table 6 shows that the Mixed Loss and coordinate attention mechanism provide significant improvement in all evaluation metrics of the PrivateLT segmentation task. Among them, compared with the benchmark model PSPNet, the network with the Mixed Loss improved its mIoU, mPA, Precision and Recall by 1.29%, 0.23%, 2.7% and 0.23%. Incorporating the coordinate attention module into the baseline model, the evaluation metric mIoU, mPA, Precision and Recall are measured, and improved by 2.65%, 0.36%, 3.15% and 0.36%, indicating that the adopted coordinate attention module enables the model to acquire the location of the tumor in order to enhance the segmentation accuracy. Finally, applying the above two strategies to PSPNet, all the metrics have been greatly improved, with mIoU, mPA, Precision and Recall improved by 3.08%, 0.32%, 3.74% and 0.32%, respectively.

In conclusion, compared to the original PSPNet network, the segmentation results with both the Mixed Loss and coordinate attention are somewhat improved, making the final segmentation results closer to the labeled graph. The effectiveness of adding a hybrid loss function and coordinate attention on segmentation in a PSPNet network is proved.

#### 4.4.6. Effect of Loss Function on Segmentation Accuracy

To test the degree of influence of the adopted Mixed Loss on the segmentation accuracy, the magnitude of the trade-off factor λ is discussed in this paper. First, a step size of 0.1 is used to obtain the value of λ, and the tested mIOU results are obtained as shown in Figure 7, where it can be seen that the trade-off factor λ takes on different values, and the obtained values of the mIOU are also different. When λ takes the value of 0, the result of mIOU is only 69.6%, and with the gradual increase in λ value, the value of mIOU rises rapidly, and the segmentation result reaches the maximum value of 71.40% when the value of λ is selected as 0.6. By examining the impact of the λ value on the segmentation results, we can assess its influence. Finally, the λ value is set to 0.6.

#### 4.4.7. Comparison of Number of Parameters in Different Networks

In order to verify that the proposed network has the advantages of saving computational resources, the parametric counting for the model is carried out, and the parametric counting experiments were carried out on CAML-PSPNet and other networks, with the statistical results shown in Figure 8. In comparison with the Deeplabv3, HrNet, UNet and PSPNet networks, the proposed network CAML-PSPNet has only 2.39 M parameters, without this affecting the segmentation accuracy. To further verify that using MobileNetV2 as the backbone network can significantly reduce the number of parameters of the proposed model, Figure 9 shows the number of parameters of the proposed model when the backbone network is replaced with ResNet-34, ResNet-50, and MobileNetV2, respectively. The results indicate that the use of MobilenetV2 as the backbone network of the model is able to reduce the count of parameters of the model effectively and save a lot of computational resources.

#### 4.4.8. Limitations and Future Prospects

During our research, we had limited access to high-performance computing resources. This restricted the scale of experiments we could conduct, especially when training more complex models. As a result, our model may not be as optimized as it could be. We have recently secured access to a more powerful computing cluster. This will enable us to train more advanced models, perform more extensive hyperparameter tuning and conduct in-depth sensitivity analyses.

#### 4.4.9. Discussion

In the research of medical image segmentation, errors caused by the model structure occurred. We found that for some image regions with complex textures and blurred boundaries, the model’s segmentation performance was not satisfactory. Through analysis, this was due to the limited receptive field of the convolutional layers in the model, which was unable to fully capture the global information of these regions. Meanwhile, we tested the model on three different medical image datasets. The PrivateLT dataset contains a large number of high-resolution lung CT images, the Kvasir-SEG dataset consists of low-resolution Colonoscopy images and the ISIC 2017 dataset is composed of dermatoscope images that are susceptible to external influences. The test results showed that on the PrivateLT dataset, the average error rate of the model was 5%, and the main errors were concentrated in the missed detection of small lung lesions. On the Kvasir-SEG dataset, the error rate increased to 7%. Due to the low image resolution, the model inaccurately judged the size and location of rectal lesion areas. On the ISIC 2017 dataset, the error rate was as high as 11%. This was because the surface of dermatoscope images was easily affected by factors such as hair and oil, and the lesion shapes were diverse, resulting in more errors during the model’s segmentation. Through such comparisons, we can more clearly understand the impact of factors such as data resolution, noise level and image type on the model error.

In the comparative experiments, we selected two networks, Swin-UNETR and TransUNet, which have emerged in recent years, for comparative experiments. The results show that the mIOU of these two networks on the datasets PrivateLT, Kvasir-SEG and ISIC 2017 is 63.72%, 71.60%, and 80.56% and 65.22%, 73.55%, and 81.20%, respectively, which is significantly lower than the segmentation results of the proposed network. The segmentation Precision is also much lower than that of the proposed network. Meanwhile, it can be seen from the visual diagrams of the segmentation results of different networks that there are still certain differences between the segmentation of the target by the Swin-UNETR and TransUNet networks and the label values. We believe that the computational resource limitations have affected the experimental results to some extent. The transformer frameworks adopted by the Swin-UNETR and TransUNet networks often require a sufficient amount of data and a large number of training epochs. However, this will increase the training time and there is a problem of training interruption. This requires higher-performance GPU resources and larger-capacity memory resources.

## 5. Conclusions

This paper investigates the application of CAML-PSPNet in medical image segmentation tasks. Aiming at the problems that usually exist for medical images, such as the problem of missed segmentation caused by a small target region and the low segmentation accuracy caused by the fuzzy boundary of the target, which is difficult to distinguish, this study proposes a method based on a CAML-PSPNet network. This model stands to significantly enhance segmentation performance, offering a robust tool for doctors in the interpretation of complex imaging data and ultimately contributing to improved diagnostic accuracy.

CAML-PSPNet solves the medical image segmentation task by introducing a coordinate attention mechanism and Mixed Loss, a hybrid loss function. Firstly, in the mixed loss function, Dice Loss serves as a metric to quantify the resemblance between the predicted outcomes and the actual labels to make the model more convergent. Secondly, the coordinate attention mechanism makes the network place greater emphasis on the characteristics of the target region. By fusing the position coordinate information with the input features, the model has the capability to assign varying attention weights to pixels located in different positions. In addition, to minimize computational expenses and conserve computational resources, the experiments utilize MobilenetV2 as the backbone, which has the characteristics of being lightweight and a high computational efficiency, which further boosts the network’s performance.

To validate the efficacy of CAML-PSPNet, three sets of experiments were carried out. First, CAML-PSPNet was compared with the existing Deeplabv3, HrNet, UNet and PSPNet methods on the private dataset of PrivateLT; the results show that the method is able to effectively solve the problems of missed segmentation caused by the small tumor region and inaccurate segmentation caused by the blurring of tumor boundaries. Then, comparison experiments were performed on two publicly accessible datasets, Kvasir-SEG rectal polyps and ISIC 2017 dermatology, and the results demonstrate that the method still achieves effective segmentation results on these two datasets and is significantly higher than the segmentation accuracies of other networks, which proves the proposed method is valid.

## Figures and Tables

**Figure 1 sensors-25-01117-f001:**
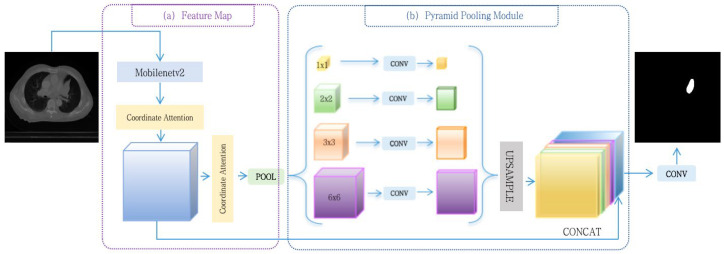
An overview of CAML-PSPNet.

**Figure 2 sensors-25-01117-f002:**
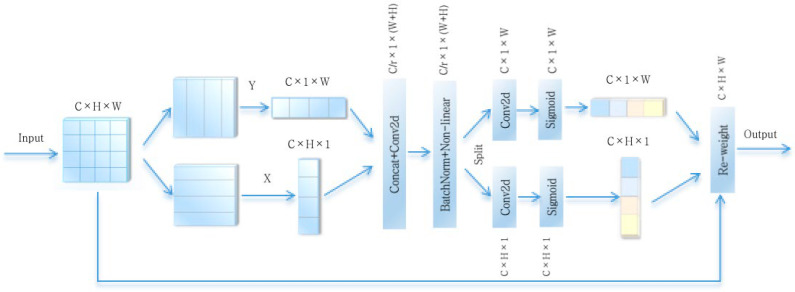
The module of coordinate attention.

**Figure 3 sensors-25-01117-f003:**
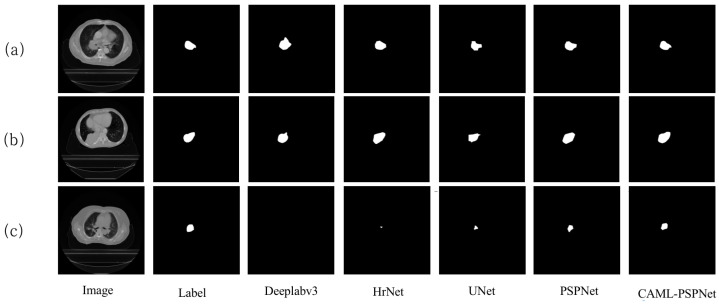
(**a**–**c**) Qualitative comparison of different methods on PrivateLT datasets.

**Figure 4 sensors-25-01117-f004:**
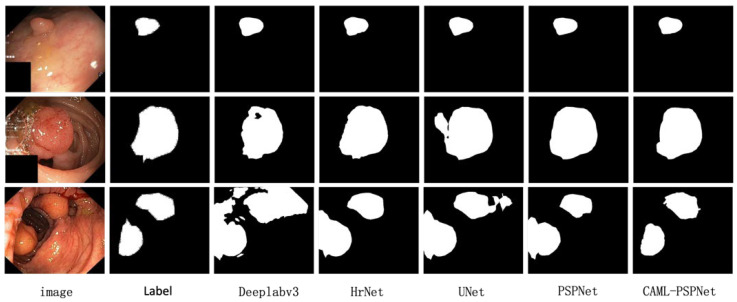
Qualitative comparison of different methods on Kvasir-SEG datasets.

**Figure 5 sensors-25-01117-f005:**
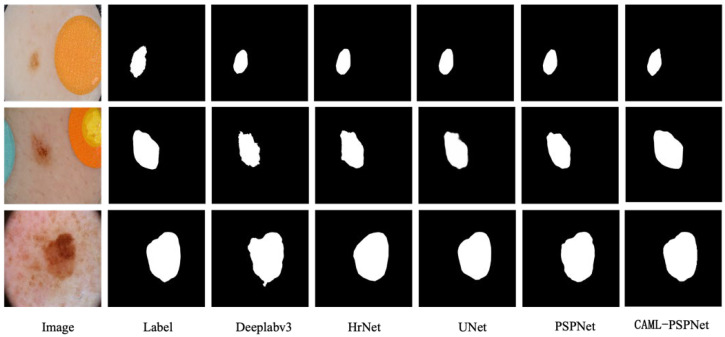
Qualitative comparison of different methods on ISIC 2017 datasets.

**Figure 6 sensors-25-01117-f006:**
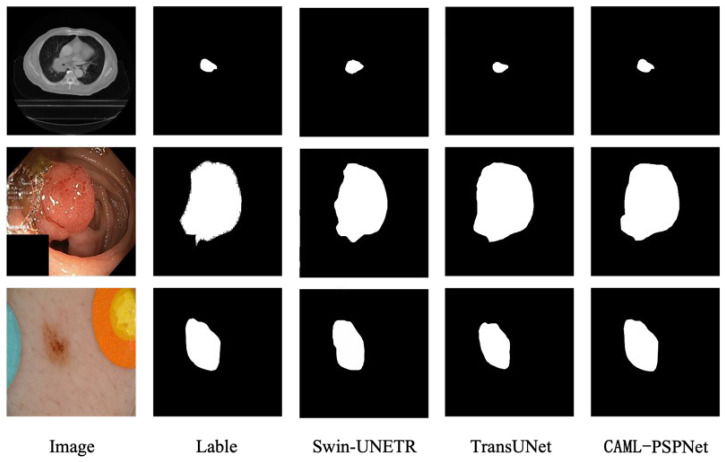
Qualitative comparison of comparative methods on three datasets.

**Figure 7 sensors-25-01117-f007:**
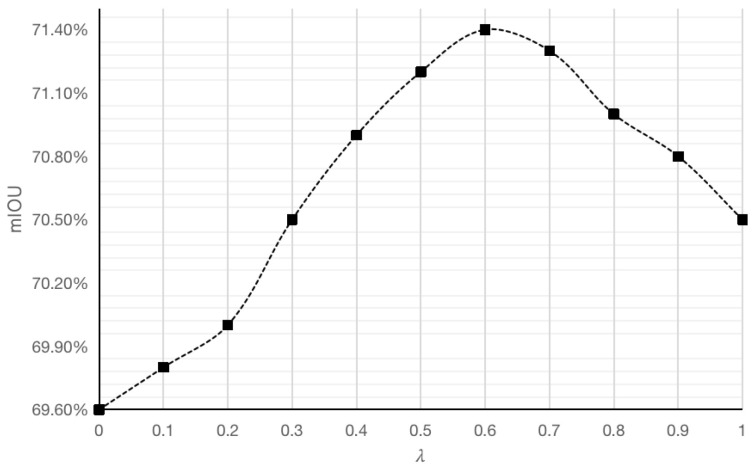
The trend in mIOU values as a function of the trade-off factor λ.

**Figure 8 sensors-25-01117-f008:**
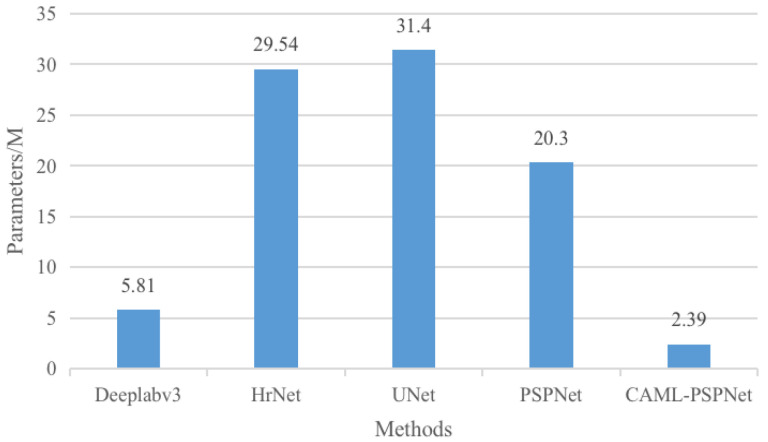
Comparison of different network parameters.

**Figure 9 sensors-25-01117-f009:**
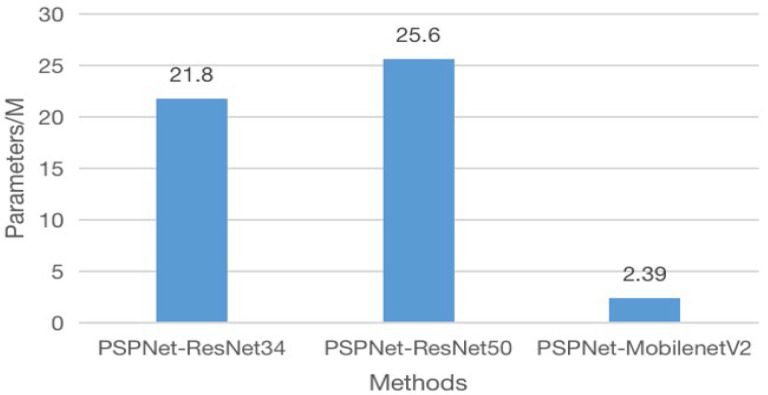
Comparison of different backbone network parameters.

**Table 1 sensors-25-01117-t001:** Dataset details.

Datasets	Modality	Number of Images	Image Size
PrivateLT	CT	1008	512 × 512
Kvasir-SEG	Colonoscopy image	1001	Different sizes
ISIC 2017	Skin lesions	960	256 × 256

**Table 2 sensors-25-01117-t002:** Segmentation results of different networks in PrivateLT datasets (unit: %).

Methods	mIOU	mPA	Precision	Recall
Deeplabv3	70.36	80.40	84.67	80.40
HrNet	70.10	80.27	84.88	80.27
UNet	67.8	79.83	81.30	79.83
PSPNet	70.12	84.87	80.13	84.87
CAML-PSPNet	73.20	85.19	83.87	85.19

**Table 3 sensors-25-01117-t003:** Segmentation results of different networks in Kvasir-SEG datasets (unit: %).

Methods	mIOU	mPA	Precision	Recall
Deeplabv3	75.77	84.77	87.71	84.77
HrNet	80.01	86.07	91.91	86.07
UNet	77.40	83.24	90.66	83.24
PSPNet	74.53	81.18	90.10	81.18
CAML-PSPNet	83.31	90.14	91.67	90.14

**Table 4 sensors-25-01117-t004:** Segmentation results of different networks in ISIC 2017 datasets (unit: %).

Methods	mIOU	mPA	Precision	Recall
Deeplabv3	86.16	93.51	91.64	93.51
HrNet	87.42	92.70	93.88	92.70
UNet	84.30	90.66	91.00	90.66
PSPNet	87.35	93.39	93.11	93.39
CAML-PSPNet	88.13	94.05	93.89	94.05

**Table 5 sensors-25-01117-t005:** Segmentation results of three datasets for compared networks Swin-UNETR and TransUNet (unit: %).

	Datasets	PrivateLT		Kvasir-SEG		ISIC 2017	

Methods		mIOU	Precision	mIOU	Precision	mIOU	Precision
Swin-UNETR		63.72	70.10	71.60	82.71	80.56	85.44
TransUNet		65.22	76.35	73.55	85.90	81.20	87.30
CAML-PSPNet		73.20	83.87	83.31	91.67	88.13	93.89

**Table 6 sensors-25-01117-t006:** Result of ablation experiments (unit: %).

Methods	mIOU	mPA	Precision	Recall
PSPNet	70.12	84.87	80.13	84.87
PSPNet+Mixed Loss	71.41	85.10	82.83	85.10
PSPNet+Att	72.77	85.23	83.28	85.23
CAML-PSPNet	73.20	85.19	83.87	85.19

## Data Availability

PrivateLT data are available from the corresponding authors on reasonable request, and other data underlying the results presented in this paper are available in Refs. [36,37].

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
