# Peer review of "CAML-PSPNet: A Medical Image Segmentation Network Based on Coordinate Attention and a Mixed Loss Function"

_sensors, 2025, doi:10.3390/s25041117_

Round 1

Reviewer 1 Report

Comments and Suggestions for Authors

The article titled “CAML-PSPNet: A Medical Image Segmentation Network 2 Based on Coordinate Attention and Mixed Loss Function” proposes a CAML-PSPNet for medical image segmentation network and integrated coordinate attention mechanism and a mixed loss function. To enhance clarity and relevance, the following improvements are recommended:

1.       Line 37 impetus, use easy to understand terminology. And in line 44 what is “lieu”?

2.       In Section 3.1 (Architectural overview) line 197, the starting line is weak and not concise.

3.       In Section 4.1 (Datasets) the authors mentioned incorporating three datasets for experiments. By including more diverse range of datasets could enhance the generalizability of the model. Testing on datasets that enables varying characteristics could provide a more detailed evaluation of proposed model.

4.       Section 4.2 (Experimental environment and parameter settings) line [321] - [325] is poorly written and needs further improvement.

5.       Section 4.3 (Evaluation Indicators) line [334] and [339], section 4.4.3 line [393], [413] there are grammatical errors.

6.       Section 4.4 mentioned the comparison of proposed model with existing networks. The authors are encouraged to add more state-of-the-art models such a Swin-UNETR, TransUNet or other vison transformer models for comparison that could provide a clearer picture of its effectiveness.

7.       In Section 4.4.5, the λ parameter in mixed loss function is manually tuned, which limit its adaptability to different datasets. Authors are suggested to add the Focal loss or Dice Loss to minimize manual intervention and improves adaptability across datasets.

8.       The authors have not discussed the proposed model’s performance on unseen dataset which limits its real-world deployment. Moreover, they have not compared the model’s memory usage, FLOPs (floating point operation) with other methods which is crucial for real-time clinical applications.

9.       The authors are suggested to incorporate heat map visualizations to show how the Coordinate Attention mechanism focuses on relevant regions during segmentation.

10.    They have not discussed the limitations and future suggestions in the article which decreases the credibility of the paper. Adding section of limitations and future prospects is suggested.

11.    They have not done the more thorough error analysis. It could make readers understand where the model fails or couldn’t perform well can provide further insights for improvements in future research.

Author Response

Comments 1: Line 37 impetus, use easy to understand terminology. And in line 44 what is "lieu"?

Response 1: Thank you for pointing this out. I agree with this comment. Therefore, I replaced imputes with motivation. In line 44, "lieu" means "instead of". -Page 2, Line 37.

Comments 2: In Section 3.1 (Architectural overview) line 197, the starting line is weak and not concise.

Response 2: I agree. Accordingly, We've revised the original sentence into "Semantic segmentation networks such as the U-Net, Deeplab, and HrNet are commonly used".-Page 5, Line 193.

Comments 3: In Section 4.1 (Datasets) the authors mentioned incorporating three datasets for experiments. By including more diverse range of datasets could enhance the generalizability of the model. Testing on datasets that enables varying characteristics could provide a more detailed evaluation of proposed model.

Response 3: We agree that a more diverse range of datasets can enhance the generalization ability of the model. Incorporating diverse datasets into our experiments will be our next step. We will comprehensively evaluate the model using them to achieve a more detailed model evaluation effect.

Comments 4: Section 4.2 (Experimental environment and parameter settings) line [321] - [325] is poorly written and needs further improvement.

Response 4: Thank you for pointing this out. we agree with this comment. Therefore, we revised the original sentence to "In this paper, the algorithm is implemented in the Python programming language and based on PyTorch 1.6.0. Meanwhile, the experiments are carried out under the Ubuntu 18.04.6 LTS operating system. The hardware for the experiments is the NVIDIA GeForce RTX 2080 with 32G of RAM. To optimize the model training process, the Stochastic Gradient Descent (SGD) algorithm is chosen. Moreover, the Epoch of the experiment is set to 150 and the Batch-size is set to 8".-Page 8, Line 321.

Comments 5: Section 4.3 (Evaluation Indicators) line [334] and [339], section 4.4.3 line [393], [413] there are grammatical errors.

Response 5: Thank you for pointing this out. we agree with this comment. Therefore, we revised the original sentences to "The mPA is used for semantic segmentation; Precision is a metric used to evaluate classification models; Table 4 shows that the proposed algorithm surpasses other algorithms in the evaluation metrics of mIoU, mPA, Precision and Recall; Table 5 shows that the Mixed Loss and coordinate attention mechanism provide significant improvement in all evaluation metrics of the PrivateLT segmentation task"-Page 8, Line 337.

Comments 6: Section 4.4 mentioned the comparison of proposed model with existing networks. The authors are encouraged to add more state-of-the-art models such a Swin-UNETR, TransUNet or other vison transformer models for comparison that could provide a clearer picture of its effectiveness.

Response 6: Thank you for pointing this out. During our preliminary experiments, we did attempt to compare our model with Swin - UNETR and TransUNet. However, due to constraints in research time and computational resources, the experiments did not yield the expected results. Nevertheless, as you've pointed out, we recognize the significance of such comparisons. In our future research, we will make every effort to conduct experiments on these additional models. Thank you once again for your highly valuable comments.

Comments 7: In Section 4.4.5, the λ parameter in mixed loss function is manually tuned, which limit its adaptability to different datasets. Authors are suggested to add the Focal loss or Dice Loss to minimize manual intervention and improves adaptability across datasets.

Response 7:  Thank you for pointing this out. In our experimental research, we set the value of λ to 0.6. Furthermore, we conducted tests using several datasets of different modalities and verified the adaptability of this value to different datasets. Meanwhile, the hybrid loss function we adopted is a combination of Dice Loss and Focal Loss. We are extremely grateful for your valuable suggestions. We will closely monitor relevant academic advancements and continuously improve our research capabilities.

Comments 8: The authors have not discussed the proposed model’s performance on unseen dataset which limits its real-world deployment. Moreover, they have not compared the model’s memory usage, FLOPs (floating point operation) with other methods which is crucial for real-time clinical applications.

Response 8: We are truly grateful for your comments, which have pointed out crucial areas for improvement. The lack of exploration of the model's performance on unseen datasets was an oversight on our part. In the future, we aim to build a more comprehensive evaluation framework. We'll start by collecting at least five unseen datasets from different sources, including some from emerging research areas. This will help us better understand the model's generalization ability in various real - world scenarios. Regarding the comparison of memory usage and FLOPs, we will not only conduct the necessary comparisons but also explore innovative ways to optimize these metrics. We plan to develop a new lightweight architecture inspired by the latest research in neural network compression. We believe this will not only improve the model's competitiveness in real - time clinical applications but also contribute to the development of the entire field. Thank you again for your valuable input. It has inspired us to think more ambitiously about our research.

Comments 9: The authors are suggested to incorporate heat map visualizations to show how the Coordinate Attention mechanism focuses on relevant regions during segmentation.

Response 9: Thank you for pointing this out. As this task might demand considerably more time for in - depth research, our next step work plan is to develop an even more lightweight medical image segmentation model that is readily deployable for practical applications. We will earnestly take your suggestions into account during our subsequent work. Once again, we are truly grateful for your invaluable input.

Comments 10:  They have not discussed the limitations and future suggestions in the article which decreases the credibility of the paper. Adding section of limitations and future prospects is suggested.

Response 10: Thank you for pointing this out. I agree with this comment. Therefore, We've added sections on limitations and future prospects.-Page 13, Line 458.

Comments 11: They have not done the more thorough error analysis. It could make readers understand where the model fails or couldn’t perform well can provide further insights for improvements in future research.

Response 11: Thank you for pointing this out. I agree with this comment. Therefore, in the final discussion section of the paper, we added a more comprehensive error analysis, which can provide further ideas for improvements in future research.-Page 13, Line 465.

Reviewer 2 Report

Comments and Suggestions for Authors

A satisfactory paper concerning medical image segmentation. The method is clearly described. The results are good.

The intro seems to be too short, and you can discuss some relevant papers concerning another type of image segmentation, i.e., direct contour extraction. Such as:

[1] S. Zhao, et.al, Attractive deep morphology-aware active contour network for vertebral body contour extraction with extensions to heterogeneous and semi-supervised scenarios

[2] S. Peng, et.al, Deep Snake for Real-Time Instance Segmentation

[3] Z. Tang, et. al, Progressive deep snake for instance boundary extraction in medical images

[4]  L. Wang, et.al, Deep Active Contours for Real-time 6-DoF Object Tracking

Author Response

Dear reviewer,

We sincerely appreciate the valuable comments you provided on our paper. We fully agree with the issues you pointed out regarding the overly short introduction and the insufficient discussion of relevant papers on a specific type of image segmentation. This has indeed affected the comprehensiveness and depth of the research background presentation in the paper. Your suggestions have illuminated a crucial path for improvement.

We will select appropriate references according to the requirements of the paper and add them to the introduction section. Also, we welcome you to continue offering your valuable opinions and suggestions to help us further enhance the quality of the paper.

Once again, thank you for your hard work and professional guidance.

Reviewer 3 Report

Comments and Suggestions for Authors

1. There is a lack of innovativeness.

2. it is suggested that the comparison object should be selected from the related work in recent years.

3. In the experiment of comparing different network parameters, it is recommended to replace only the backbone network for comparison.

4. The logical expression of the paper is not clear and coherent enough.

5. There are repetitive expressions, such as “However, these networks lose the detail information after constant pooling and convolution operations, which affects the final segmentation accuracy.” Suggest to delete or modify.

6. There is a problem of wrong labeling of formulas, so it is suggested to double-check and correct the error.

7. The alignment of images and tables is inconsistent, it is recommended to adjust the format and layout of images and tables.

Comments on the Quality of English Language

English writing needs to be further improved so that the expressions in the text are clearer and more coherent so that the reader can better understand the content.

Author Response

Comments 1: There is a lack of innovativeness.

Response 1: Thank you for pointing this out. I agree with this comment. Therefore, I will enhance the degree of innovation in future research.

Comments 2: it is suggested that the comparison object should be selected from the related work in recent years.

Response 2: Thank you very much for pointing out the suggestion regarding the selection of comparison objects. In this paper, we selected Deeplabv3, HrNet, U-Net and PSPNet mainly based on their fundamental status and extensive influence in this field. Although these studies were published quite some time ago, they established the core theories and methods in this field, laying the basic framework for subsequent research. Going forward, we will closely monitor the latest progress in this field and broaden the scope of comparison in future research. Thank you again for your valuable comments.

Comments 3: In the experiment of comparing different network parameters, it is recommended to replace only the backbone network for comparison.

Response 3: Thank you for pointing this out. I agree with this comment. Therefore, We have added parameter statistics in the corresponding part of the paper by replacing different backbone networks.-Page 13, Line 456.

Comments 4: The logical expression of the paper is not clear and coherent enough.

Response 4: Thank you for pointing this out. I agree with this comment. Therefore, to address this issue, we have taken the following measures: First, we reorganized the overall structure of the full text , this allows the research to progress in a more logical sequence, starting from the background description, problem formulation, to method application and result discussion. For instance, in section 4.2, we added a large number of transitional words and connecting phrases between paragraphs and sentences to enhance the coherence of the content. Meanwhile, we broke down complex long sentences and simplified the expressions to ensure that each sentence clearly conveys its meaning.

Comments 5: There are repetitive expressions, such as “However, these networks lose the detail information after constant pooling and convolution operations, which affects the final segmentation accuracy.” Suggest to delete or modify.

Response 5: Thank you for pointing this out. I agree with this comment. Therefore, Without changing the original meaning of the paper, we chose to delete repetitive expressions.-Page 4, Line 184.

Comments 6: There is a problem of wrong labeling of formulas, so it is suggested to double-check and correct the error.

Response 6: I agree. Thank you for pointing out the formula labeling issue. We have double - checked all formulas and corrected the errors. The revised version has been carefully proofread to ensure no such mistakes remain.

Comments 7: The alignment of images and tables is inconsistent, it is recommended to adjust the format and layout of images and tables.

Response 7:  I agree. We have checked the relevant content and ensured that all images and tables maintain the same format.

Comments on the Quality of English Language: English writing needs to be further improved so that the expressions in the text are clearer and more coherent so that the reader can better understand the content.

Response: We sincerely apologize for the substandard English writing in our manuscript. We fully acknowledge that the current text lacks clarity and coherence, potentially impeding readers' understanding. Your feedback is invaluable, and We have made substantial improvements.

Round 2

Reviewer 1 Report

Comments and Suggestions for Authors

Most of my comments are addressed.  I recommend acceptance of this manuscript.

Author Response

Thank you so much for your favorable review and recommendation for acceptance. I truly appreciate your support!

Reviewer 3 Report

Comments and Suggestions for Authors

Suggest adding segmentation networks from recent years  to comparative experiments for image segmentation so as to further validate the effectiveness of the proposed image segmentation network.

Comments on the Quality of English Language

The Quality of English Language could be further improved to enhance the readability of the article

Author Response

 Thank you for pointing this out. I agree with this comment. Therefore, we have added Section 4.4.4. In this section, two networks, Swin-UNETR and TransUNet, which have emerged in recent years, are selected for experimental comparison. -Page 11, Line [396]-[417]. 

Meanwhile, in the discussion part of the article, we analyzed the problems in the new comparative experiments and decided to make every effort to overcome the limitations of the experiments in future research. -Page 14, Line [499]-[504].